# Psychological Factors Influencing Adherence to NIV in Neuromuscular Patients Dependent on Non Invasive Mechanical Ventilation: Preliminary Results

**DOI:** 10.3390/jcm12185866

**Published:** 2023-09-09

**Authors:** Anna Annunziata, Cecilia Calabrese, Francesca Simioli, Antonietta Coppola, Paola Pierucci, Domenica Francesca Mariniello, Giuseppe Fiorentino

**Affiliations:** 1Unit of Respiratory Pathophysiology, Critic Area Department, Monaldi—Cotugno Hospital, 80131 Naples, Italy; anna.annunziata@gmail.com (A.A.); francesimioli@gmail.com (F.S.); giuseppefiorentino1@gmail.com (G.F.); 2Department of Translational Medical Sciences, University of Campania “Luigi Vanvitelli”, 80138 Naples, Italy; ceciliacalabrese123@gmail.com (C.C.); domenicafrancemariniello@gmail.com (D.F.M.); 3Cardiothoracic Department, Respiratory and Critical Care Unit, Bari Policlinic University Hospital, 70121 Bari, Italy; paola.pierucci@policlinico.ba.it; 4Section of Respiratory Diseases, Department of Basic Medical Science Neuroscience and Sense Organs, University of Bari ‘Aldo Moro’, 70122 Bari, Italy

**Keywords:** anxiety, NIV acceptance, non-invasive mechanical ventilation

## Abstract

Background: Non-invasive ventilation (NIV) is associated with improvement of both morbility and mortality in patients affected by neuromuscular diseases with chronic respiratory failure. Several studies have also shown that long-term NIV positively impacts the patient’s quality of life and perception of disease status. Its effectiveness is likely related to the adherence to NIV. Several factors, patient- and not patient-related, may compromise adherence to NIV, such as physical, behavioral, familiar, and social issues. Few data are currently available on the role of psychological factors in influencing NIV adherence. Materials and methods: In this pilot study, we evaluated the adherence to NIV in a group of 15 adult patients with neuromuscular diseases (Duchenne muscular dystrophy, myotonic dystrophy, and amyotrophic lateral sclerosis) in relation to their grade of depression assessed by the Beck Depression Inventory (BDI) questionnaire. Other data were collected, such as clinical features (age and sex), use of anxiolytic drugs, the presence of a family or professional caregiver, the quality of patient–physician relationship, the beginning of psychological support after BDI screening, and the family acceptance of NIV. NIV adherence was definied as the use of NIV for at least 4 h per night on 70% of nights in a month. Results: The overall rate of NIV adherence was 60%. Based on the BDI questionnaire, patients who were non-adherent to NIV had a higher rate of depression, mainly observed in the oldest patients. The acceptance of NIV by the family and positive physician–patient interaction seem to favor NIV adherence. Conclusion: Depression can interfere with NIV adherence in patients with neuromuscolar diseases.

## 1. Introduction

Non-invasive ventilation (NIV) is routinely prescribed in patients affected by numerous neuromuscular diseases with acute or chronic respiratory failure in order to support respiratory muscle weakness and improve gas exchange. NIV is associated with an improvement of both morbility and mortality and several studies have also shown that long-term NIV positively impacts the patient’s quality of life and perception of disease status [1,2]. In contrast, non-acceptance or incorrect use of NIV often results in worse clinical outcomes and greater healthcare costs [3,4]. However, NIV effectiveness is likely related to its adherence. Several studies have shown that NIV adherence is often poor in patients with neuromuscular diseases. Rautemaa et al. found that 72% of patients with type 1 myotonic dystrophy scarcely used NIV (<5 h per 24 h) [5], although a recent multicenter retrospective analysis demonstrated that NIV was used more than 70% of nights with an average nightly usage of 7 h by patients with Duchenne muscular dystrophy, particularly by older subjects [6].

During COVID-19 pandemic waves, the serious impacts of undergoing an urgent tracheotomy for patients affected by neuromuscolar diseases due to poor NIV compliance and/or lack of clinical follow-up have been demonstrated to have an heavy impact on both patients’ and caregivers’ mental health [7,8]. Several factors may influence adherence to NIV. For example, severity of disease, perception of benefits, a healthy style of life, and psychological factors, such as fear of tracheostomy and/or death, have been reported to positively influence NIV adherence [9]. In the same way, the presence of a professional or familiar caregiver, as well as a positive doctor–patient relationship, can favor the adherence to non-invasive support [10].

Though NIV can significantly improve clinical symptoms, patients often reject NIV or do not use it appropriately. Reasons for refusal or withdrawal of NIV are not only related to the treatment itself but also to psychological aspects of neuromuscolar patients. Generally, problems with the interfaces and the exposure to forced air pressure represent the main causes for non-acceptance of NIV [11]. Choosing the right interface may have a huge impact on NIV effectiveness, duration, and compliance [12]. However, the aderence to NIV may vary over time: in fact, the clinical progression of the disease implies an increase in the overall time of NIV use that further increases the risk of pressure ulcers, decubitus, and gastric distension. Using alternative NIV supplies or strategies may promote adherence in patients requiring long-term NIV [13]. A previous “case report” described a patient who initially refused NIV but subsequently accepted it when a less “cumbersome” interface was adopted [14]. Intermittent abdominal pressure ventilation, working with an abdominal compression in the absence of an interface, should be considered in all cases of intolerance to NIV masks for problems of decubitus, claustrophobia, or visual field limitation [15]. Mouthpiece ventilation, through a simple angle mouthpiece, may favor the acceptance of NIV [16]. Negative pressure ventilation, delivering pression by cuirass or poncho applied on the thorax, provides a pattern of breathing similar to spontaneous breathing [13]. In addition, patients with chronic neuromuscolar diseases requiring NIV can experience a psychological impairment that, in turn, can aggravate their symptoms, the course of the disease, and the adherence to NIV [10]. In fact, depression and anxiety in neuromuscolar patients have been reported to increase the episodes of dyspnea and the number of hospitalizations due to exacerbations and the risk of death, and to worsen health-related quality of life [17,18]. In addition, in patients requiring long-term NIV, the depencence on ventilatory support further compromises the psychological status of the patient, since his autonomy and social life are severely limited as compared to patients who need intermittent NIV. Consequently, the emotional burden can be responsible for an inappropriate adherence to NIV. On the contrary, Chao et al. observed that the overall time of use of NIV increases during the disease progression; NIV adherence in advanced stages is more likely associated to worsening of symptoms such as dyspnea and fatigue, usually due to the decrease of vital capacity [19].

In order to improve NIV adherence, the identification of modifiable predictors of poor acceptance of NIV is necessary to promote interventions targeting adherence barriers. Few data are currently available on the role of psychological factors in influencing NIV adherence, although several studies have demonstrated the presence of emotional disorders in neuromuscolar patients [20]. For example, in a study by Simmons et al., the authors found that psychological and existential factors are better predictors of quality of life than physical impairment in patients with amyotrophic lateral sclerosis [21].

A recent study evaluating the psychosocial attitude of young boys with Duchenne muscular dystrophy showed that these patients do not have a significant increase in internalizing (e.g., anxiety, depression) or externalizing problems when compared to age-matched patients, except for higher obsessive-compulsive tendencies. Moreover, the study suggests that a better parental support and an increased family involvement reduce their behavioral symptoms. In contrast, family stress may have a significant impact on depression and psychosocial functioning in these patients [22]. In a study on patients with type 1 myotonic dystrophy, signs of clinical depression were identified in 32% of patients, and the depressive condition was mild to moderate and comparable to ratings in other neuromuscular diseases [23]. A large population-based study demonstrated a higher risk of clinical depression in patients affected by amyotrophic lateral sclerosis and their use of antidepressant medications immediately before and after the neuromuscular diagnosis. The highly increased risk of depression during the first year after the diagnosis of the disease may reflect a severe psychological reaction to the neuromuscular disease diagnosis. Whether depression should be considered a prodromal symptom of amyotrophic lateral sclerosis remains to be elucidated [24]. Unfortunately, clinicians are often unaware of the emotional and/or behavioral problems of neuromuscular patients and how these factors influence NIV adherence and consequently their clinical outcomes. For these reasons, we investigated the impact of depressive symptoms on NIV adherence in patients affected by neuromuscular diseases with chronic respiratory failure.

## 2. Methods

In this pilot study, we investigated the presence of depressive symptoms in a small cohort of patients affected by chronic neuromuscular disorders and candidates to NIV, followed at the Unit of Respiratory Pathophysiology, Critic Area Department, Monaldi–Cotugno Hospital, Naples, Italy in a period between March and September 2022.

A validated Italian version of the Beck Depression Inventory (BDI) questionnaire was admnistered to all neuromuscular diseases patients following the diagnosis of chronic respiratory failure, before starting the trial with NIV [25]. The BDI is a self-report screening tool to assess both cognitive and somatic aspects of depression, composed of 21 items that investigate the presence and the degree of depressive symptoms, such as hopelessness and irritability, cognitions such as guilt or feelings of being punished, and physical symptoms such as fatigue, weight loss, and lack of interest in sex. Each of the 21 items is rated from 0 to 3, with a total score ranging from 0 to 63. A total score under 10 is considered normal, a score between 11 and 16 indicates a mild mood disturbance, 17 and 20 a borderline clinical depression, 21 and 30 a moderate depression, 31 and 40 a severe depression, and over 40 an extreme depression.

Clinical features (age and sex), the type of neuromuscular disease, the BDI score, the use of anxiolytic drugs, the presence of a family or professional caregiver, the self-reported quality of the relationship with the own medical doctor, the family acceptance of NIV, the NIV adherence and the presence of psychological support after BDI screening were assessed (Table 1).

Adherence was defined as the use of NIV for at least 4 h per night on 70% of nights within a month. Although the minimal time to therapeutic effect has not been clearly stated in literature, this is a common clinical assumption based on empirical observations [19,26].

## 3. Results

A total of 15 patients were recruited: 3 patients affected by Duchenne muscular dystrophy, 3 by myotonic dystrophy, and 9 by amyotrophic lateral sclerosis. The median age of patients was 52 years [IQR 30–64]. The majority of patients were males (8 patients, 53%). The median BDI score was 17 [IQR 11–24], indicative of a borderline clinical depression. All patients had a caregiver: in 13 cases (87%), there was a familiar careviger, while 2 patients (13%) had a professional caregiver (cases 6 and 13). A total of 11 patients (73%) reported a good physician–patient relationship. NIV usage was accepted by family in 10 patients (67%).

NIV adherence was observed in 9 patients (60%). In this subgroup of patients, the median age was 52 years [IQR 33–70] and the median BDI score was 14 [IQR 9–19], indicative of a mild mood disturbance. Patients adherent to NIV stated that they believe in the utility of the ventilatory support with a good relief of their dyspnea. In all cases of NIV adherence, there was a family acceptance, so we can argue a beneficial role of the familiar support. Similary, all patients adherent to NIV reported a good relationship with their physician, which can positively influence the use of the ventilatory support. A total of 6 patients (40%) showed a poor adherence to NIV. In these patients, the median age was 47 years [IQR 30–61] and the median BDI score was 23 [IQR 17–31], indicative of a moderate depression. Among them, the youngest patients (cases 4 and 9) had a mild depressive disturbance, while the oldest (cases 3, 6, 7, and 11) showed a moderate to extremely severe depression. For 5 patients (83%) who are not adherent to NIV, family did not accept NIV. Likewise, 4 patients (66%) did not report a good relationship with their medical doctor. Thus, the lack of family and physician support can lead to impaired NIV adherence. The most common reasons for non-adherence to NIV were discomfort with interfaces and non-comprehension of utility and effectiveness. In fact, 3 patients stated that they had not received adequate medical information about their disease and NIV support.

No patients received a psychiatric diagnosis prior to the administration of the BDI questionnaire, although 2 patients (cases 7 and 8) used anxiolytic therapy as needed (Diazepam 5 mg bid and Lorazepam 2.5 mg, respectively) prescribed by their family doctor.

After the BDI screening, the diagnosis of depression was made in 8 patients (53%) who had a BDI score > 17. These patients were evaluated by a psychiatrist specialist working in our department in order to support adult subjects with neuromuscular diseases. A total of 4 patients (cases 7, 8, 10, 12) agreed to start a psychological support, although one of them (case 7) interrupted it due to a clinical deterioration.

## 4. Discussion

In this descriptive study performed on a small cohort of patients affected by chronic respiratory failure secondary to three different neuromuscolar diseases (Duchenne muscular dystrophy, myotonic dystrophy, and amyotrophic lateral sclerosis) and who were candidates to perform NIV, we observed that 60% of patients were adherent to the ventilatory support, i.e., they used NIV for at least 4 h per night on 70% of nights in a month. When we assessed the score of the BDI questionnaire, we found a higher degree of depression in the non adherent NIV patients in comparison with the adherent ones. Although the small sample size of our study population does not allow us to perform a statistical analysis, our observation is, in our opinion, very interesting considering that, among factors influencing NIV adherence in neuromuscular patients, the impact of depression has been investigated in only a few studies. Pascoe et al., in a cohort of 42 youths with Duchenne muscular dystrophy, demonstrated that symptoms such as anxiety and depression are predictors of lower NIV adherence. They also showed that a single 90 min session of behavioral therapy provided by a psychologist was effective in increasing NIV usage [22]. In contrast, Russo et al., in patients affected by amyotrophic lateral sclerosis, found no correlation between anxiety and depression and the time of adaptation to NIV but they found that decreased motivation/initiation, poor mental flexibility, and difficulty in planning may be negative predictors of NIV adaptation [27]. Dorst et al. evaluated the effects of NIV on quality of sleep, daytime fatigue, and depression in patients with amyotrophic lateral sclerosis but they did not observe an improvement of score of depression after NIV initiation [28].

As mentioned above, in our study, we found a higher degree of depression in non-adherent patients in comparison with the adherent ones. In contrast with our observations, Martin et al., in a cohort of patients affected by amyotrophic lateral sclerosis, demonstrated that patients with fewer depressive symptoms at baseline were more likely to refuse an intervention such as NIV probably due to their wish to not prolong their own life using an intervention such as NIV, which, in some patients’ opinion, can undermine the sense of identity, dignity, and/or autonomy. In addition, patients who understood their illness less well and those with a proactive attitude to decision-making were more likely to refuse NIV [9].

In our study, among the non-NIV adherent patients, the highest BDI scores suggestive of a moderate/severe depression were reported by the oldest patients affected by amyotrophic lateral sclerosis. It can be hypothesized that the perception of disability by a patient with a later onset of a neuromuscular disease is probably different from that of a patient suffering from a congenital muscular disease. The former has to face a progressive change in his lifestyle and, frequently, has to give up his habits, hobbies, and passions that contribute to his psychological harmony. The need to adapt to these changes can cause an imbalance in the patient’s psychological structure and, for this reason, whenever possible, clinicians should preserve the small positive habits or hobbies that bring satisfaction. Finally, the different life expectancies between a young and an old patient can decisively interfere with NIV acceptance and adherence. Children, due to the precocity of the disability or the presence of the caregiver, are probably more inclined to change their lifestyle to NIV. Some studies point out that children with neuromuscular diseases and their families experienced a low quality of life and mental health. However, there was no additional negative influence on the overall quality of life by ventilator use [29].

In our study, all patients adherent to NIV reported a good relationship with their medical doctors, while four non-adherent patients of six did not have good interaction with their medical team, complaining of a lack of comunication and education abilities. As previously demonstrated, low quality physician–patient interactions can impair the emotional responses of a patient, in accordance with our study in which these patients had the highest scores on the BDI questionnaire. In fact, the extent to which clinicians are compassionate, receptive, and accurate in their counselling about NIV can significantly influence the patient’s perspective, which is frequently associated with a sense of threatened autonomy. Acceptance of NIV can be improved when patients feel that clinicians care about their emotional and physical needs and when the continuity of medical care by the same clinician is guaranteed [9,30]. Sometimes, patients report that they have perceived doctors enforcing NIV and more research is needed to understand the attitudes that clinicians should have in order to help patients and their families to cope with NIV.

In the present study, in all cases of adherence to NIV, there was also a family acceptance of NIV. In contrast, in patients non-ahderent to NIV, the family did not accept ventilation support. NIV adherence may be impacted when the family has limited knowledge or understanding with regard to medical illness and treatment [30]. As previously demonstrated, the role of family support is fundamental to encourage the compliance of neuromuscular patients with non-invasive support for home ventilation [31,32]. The family network represents a positive factor that promotes the desire to live; in fact, it has been shown that married neuromuscular patients more frequently accept and use NIV and live longer than unmarried ones [33,34]. Jackson et al., in 73 patients affected by amyotrophic lateral sclerosis, observed that a single marital status as well as a lower level of education and household income < 50,000 USD negatively impact compliance to NIV [35]. Chiò et al. also found that NIV was more frequently performed by married patients, who also had a longer survival after NIV than non-married patients but the patient’s social status, as indicated by educational level, was not a factor hampering the acceptance of mechanical ventilation, probably because ventilatory interventions and home support are provided at no cost by the Italian National Health System [36].

In the present study, all NIV-adherent and non-adherent patients had a caregiver, so we cannot evaluate the role of this in influencing adherence to the ventilatory support. However, the literature oulines the crucial role of the caregiver in NIV acceptance and compliance [37]. In fact, a caregiver is a person who, not only technically helps the patient to use ventilators and supplies, eventually checking the machine’s functioning, but he is often the first person who recognizes the internalizing symptoms of the patient such as sadness, anxiety, and loneliness and consequently the psychological needs of the patient. Although NIV increases the caregivers’ burden and may negatively affect their physical function with signs of exhaustion, such as insomnia, anxiety disorders, or loss of attention, a study has shown that it does not significantly impair their well-being [14]. Considering the pivotal role played by the caregiver, they should be involved in the planning of the care of neuromuscular patients; in fact, refusal of NIV by caregivers, rarely reported, could negatively influence the patient’s choice. In addition, caregivers should receive therapeutic education training.

In the present study, no patient had received the diagnosis of depression before the admnistration of the Beck questionnaire. For this reason, it could represent a valid tool to screen the presence of this comorbidity in neuromuscolar patients. Without an appropriate medical treatment and psycological support, depression tends to take over, also considering that neuromuscular diseases are associated with increasing disability over time. Depression, with its chronic and relapsing course, can interfere with other therapies, such as the acceptance and adherence to NIV. To promote NIV tolerance, particularly in patients with underlying psychiatric disorders already present, accurately titrated sedating medications can be useful; similarly, using alternative and/or complemetary forms of NIV can be indicated in patients who require NIV for many hours, expecially during daytime [13,14,15,16].

The sexual sphere also has significant repercussions on the patient’s psychology. When compared with normal subjects, the sexual activity and masturbation rate of patients with NIV are significantly reduced and they reduce further with increasing age. Eighty percent of patients attribute this reduction to the somatic effects of their chronic disease. In fact, sexual activity causes an increase in tidal volume and breathing frequency, with an increased cardiopulmonary load and exertional dyspnea [37]. Consequently, sexual activity is often reduced in subjects with chronic lung diseases, particularly in those receiving NIV for chronic respiratory failure. The decrease in sexual activity could be due not only to NIV but also to the progression of the chronic respiratory disease, which can increase fatigue and dyspnea, and, in some cases, to a depressive component associated with these medical conditions [38].

## 5. Limitations

Conclusions based on current and preliminary results are limited by several factors. A small number of patients were enrolled, so a statistycal analysis was not performed. In addition, the study population included patients with three distinct neuromuscular disorders characterized by a different clinical course. For example, the poor prognosis of amyotrophic lateral sclerosis, compared to myotonic dystrophy, could have different implications for the prevalence of a depressive disorder. Moreover, the study included patients with various ages of disease onset, which can differently impair the disability perceived by the patient and the following depressive symptoms.

Another limit of our study is that we did not investigate NIV adherence after a specific psychological support and/or medical treatment due to the fact that only four patients accepted to start it. Finally, our study did not evaluate the psychological status of the caregiver whose support can improve patient mental health and the achievement of clinical outcomes.

## 6. Conclusions

Our study outlines the impact of depressive symptoms on NIV adherence in neuromuscular patients. Family acceptance of the ventilatory support, as well as a good interaction between physician and patient, seem to favor patients’ adherence. Although there is a strong link between mental and physical conditions, which can turn into a vicious negative circle, psychological aspects, such as depressive symptoms, are often underestimated in neuromuscular patients. As shown in our study, the administration of specific questionnaires could help guide clinicians to identify the psychiatric disorder and subsequently to begin a personalized psychological support and/or medical treatment. However, no studies have evalueted whether the treatment of depressive symptoms can improve NIV compliance. The achievement of this objective is very important, considering that NIV usage could improve both morbidity and mortality as well as quality of life in patients with neuromuscular diseases.

Future prospective larger studies on a homogeneous cohort of neuromuscular patients are needed in order to investigate the impact of depressive symptoms on NIV adherence and its variation after appropriate psychiatric treatment.

## Figures and Tables

**Table 1 jcm-12-05866-t001:** Baseline characteristics of patients affected by neuromuscular diseases and their acceptance and adherence to NIV.

Case	Age	Sex	NMD	BDI Score	Anxiolytic Therapy	Family or Professional Caregiver	Good Physician-Patient Relationship	Psycological Support after BDI Screening	NIV Acceptance by Family	NIV Adherence
1	22	M	DMD	8	no	yes	yes	no	yes	yes
2	19	M	DMD	15	no	yes	yes	no	yes	yes
3	30	F	DM	21	no	yes	yes	no	no	no
4	18	M	DMD	11	no	yes	no	no	no	no
5	44	F	DM	10	no	yes	yes	no	yes	yes
6	55	F	DM	31	no	yes	no	no	yes	no
7	61	M	ALS	44	yes	yes	no	yes	no	no
8	46	M	ALS	20	yes	yes	yes	yes	yes	yes
9	39	F	ALS	17	no	yes	yes	no	no	no
10	78	F	ALS	28	no	yes	yes	yes	yes	yes
11	79	M	ALS	24	no	yes	no	no	no	no
12	64	F	ALS	19	no	yes	yes	yes	yes	yes
13	52	M	ALS	13	no	yes	yes	no	yes	yes
14	76	M	ALS	14	no	yes	yes	no	yes	yes
15	58	F	ALS	6	no	yes	yes	no	yes	yes

NMD: neuromuscolar disease; DMD: Duchenne muscular dystrophy; DM: myotonic dystrophy; ALS: amyotrophic lateral sclerosis; BDI: Beck Depression Inventory; F: female; M: male; NIV: non-invasive ventilation.

## Data Availability

No new data were created or analyzed in this study. Data sharing is not applicable to this article.

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
