# Peer review of "Psychological Factors Influencing Adherence to NIV in Neuromuscular Patients Dependent on Non Invasive Mechanical Ventilation: Preliminary Results"

_jcm, 2023, doi:10.3390/jcm12185866_

Round 1

Reviewer 1 Report

I thoroughly enjoyed reading this paper, which focuses on the factors that limit the acceptance and adherence of Non-invasive ventilation (NIV) support among patients with Neuromuscular diseases (NMD). The authors delve into the social and psychological factors impacting NIV acceptance by patients.

Strengths of the study:

The study's objective is to determine the impact of depressive symptoms on NIV compliance among patients affected by NMD. This clinically relevant and pragmatic objective greatly enhances the study's value.

The methods section is well-crafted and easy to comprehend:

·         Depression was assessed using self-reported data from a 21-item Beck Depression Inventory (BDI) questionnaire, classified on a scale ranging from mild mood disturbance to severe depression.

·         Adherence to NIV was defined as 4 hours per night for 70% of nights within a month.

The study incorporates a diverse patient population despite its small sample size, including Amyotrophic lateral sclerosis, Duchenne muscular dystrophy, and dystrophia myotonica patients.

The results are presented descriptively, including individual data for all 15 patients.

Weaknesses of the study:

The study's most prominent limitation lies in its small sample size (n=15). Although patients non-adherent to NIV exhibited higher BDI scores, the study did not perform any statistical significance tests on these results.

Minor limitations include the introduction section, which is challenging to comprehend due to its sentence structure and the inclusion of studies not directly related to this publication's objective.

Reviewer 2 Report

Dear Authors,

You choose a high specific topic for your research.

English exposure is insufficient along all your paper and there are several typing or grammatical mistakes.

Introduction is well designed and adequate for this manuscript.

Methods section missed both the timing for recruitment and there's no mention about timing between NIV start and administered BDI questionnaire.

Results are clear and considering your small sample nothing more than descriptive analysis can be done.

Discussion is too long, and some considerations are outside the objective of your research or cannot be proven.

I think that in this way, your results don't add critical innovation to the topic, even if exploring outbreak depression status in in long-term ventilated patients is crucial.

Maybe, if you could collect more data about your fifteen patients, like use of antidepressants or antipsychotics, and kind of psychological support they received during ventilation, or an external validation of psychological status by a third specialist outside the research would add more weight to your results.

English has to be improved.

There are a lot of typing and grammatical mistakes, you may control all the paper.

Round 2

Reviewer 2 Report

Dear Authors, 

Your review changed the overall presentation of your work.

This new version of the paper is greatly better than the old one.

I accept all changes you have done, in particular I appreciated the different context of some citations that looked inappropriate in the old discussion, and now complete an exhaustive introduction.

I confirm my previous opinion in methods and results.

The new version of your discussion improves the quality of your small results and correctly contextualize each factor that could, in your hypothesis, interfere with psychological approach to disease and to NIV support.

I appreciate also the extensive English revision.

Now the paper is near to publishing, I have just 2 observation:

1 - There are few typing mistakes

2 - Change "Bibliografia" with "Bibliography". 

Author Response

Thank you very much for your comments. We have provide to correct the typing mistake and change "Bibliografia" with "Bibliography"